# A Cognitive Model to Anticipate Variations of Situation Awareness and Attention for the Takeover in Highly Automated Driving

**Marlene Susanne Lisa Scharfe-Scherf** [1,2,*], **Sebastian Wiese** [2] **and Nele Russwinkel** [2]

1   Robert Bosch GmbH, Robert-Bosch-Allee 1, 74232 Abstatt, Germany
2   Department of Psychology and Ergonomics, Technical University Berlin, 10587 Berlin, Germany
*   Correspondence: marlene-susanne-lisa.scharfe@de.bosch.com

**Abstract:** The development of highly automated driving requires dynamic approaches that anticipate the cognitive state of the driver. In this paper, a cognitive model is developed that simulates a spectrum of cognitive processing and the development of situation awareness and attention guidance in different takeover situations. In order to adapt cognitive assistance systems according to individuals in different situations, it is necessary to understand and simulate dynamic processes that are performed during a takeover. To validly represent cognitive processing in a dynamic environment, the model covers different strategies of cognitive and visual processes during the takeover. To simulate the visual processing in detail, a new module for the visual attention within different traffic environments is used. The model starts with a non-driving-related task, attends the takeover request, makes an action decision and executes the corresponding action. It is evaluated against empirical data in six different driving scenarios, including three maneuvers. The interaction with different dynamic traffic scenarios that vary in their complexity is additionally represented within the model. Predictions show variances in reaction times. Furthermore, a spectrum of driving behavior in certain situations is represented and how situation awareness is gained during the takeover process. Based on such a cognitive model, an automated system could classify the driver's takeover readiness, derive the expected takeover quality and adapt the cognitive assistance for takeovers accordingly to increase safety.

**Keywords:** takeover; highly automated driving; automation; situation awareness; adaptive automation; human–automation interaction; cognitive architectures; decision making; user studies

## 1. Introduction

The development of highly automated driving is a central topic of current transportation research [1]. Still, the next generations of highly automated vehicles (Levels 3 and 4) will need the driver as fallback level [2,3]. Highly automated driving solutions thus require dynamic approaches including the anticipation of the cognitive driver state within dynamic situations and need to be adapted in a way that control could be handed over intuitively between driver and vehicle. In case the driver has to resume control, a takeover request is triggered. However, the assumption that a human driver stays alert and is always ready to takeover control within seconds is fallacious [1]. The switch between a non-driving-related task (NDRT) and the main driving task is crucial during the takeover. The driver is required to build up situation awareness according to the situation that unfolds in order to make an action decision [4,5]. Only when the driver has collected enough information of the surrounding and is in the loop [6], safe actions in a dynamic situation are possible. This process requires manual control skills as well as cognitive skills [7]. A drivers' takeover readiness when resuming control after an automated driving section may be influenced by various factors [8], inter alia cognitive and environmental factors. Within this paper, the focus lies on cognitive mechanisms and the process to build up situation awareness.

Driver behavior thereby is determined by complex interactions with other traffic participants [9] and the system itself. To increase safety and comfort, it has to be identified where humans best fit into the system [1,10].

It has been shown [11] that the usability and comprehensibility in highly automated driving can be increased when using adaptive systems. This shows that it is highly important to consider the individual driver when developing advanced driver assistance systems. For such an individual adaption in highly automated driving, the need to simulate human agents arises [12]. Thus, it becomes important to determine the takeover readiness of each individual driver and develop adaptive cognitive assistance systems to support the driver accordingly.

Such cognitive assistance systems serve the purpose of supporting the human driver during a takeover task. Such a support could be, for example, a pre-deceleration before the takeover, the best maneuver indication, or the suppression of irrelevant information in a complex situation. In the long run, these systems do not drive the vehicle, but anticipate the driver and enable the driver to takeover better and more safely. To develop such a system, cognitive processes during a takeover should be simulated and anticipated to understand the cognitive status of situation awareness within a particular situation. To illustrate cognitive processes and the development of situation awareness, this paper uses cognitive modeling to represent a spectrum of driving decisions in dependence of the current dynamic situation during a takeover.

### 1.1. Cognitive Modeling in Highly Automated Driving

The use of cognitive driver models within advanced driver assistance systems allows anticipating the driver in context to all traffic participants in the close vicinity. A prediction of other traffic participants is used by [9] to adapt the advanced driver assistance system reaction and consequently increase safety and comfort. Nevertheless, more importantly, time dynamics of driver perception, scene interpretation and decision making [12] need to be considered during the takeover to adapt assistance systems accordingly and increase safety.

A first approach to include the driver into an advanced driver assistance system is done by [8]. They propose an adaptive advanced driver assistance system to keep the driver in-the-loop or support in takeover situations through supplementary features such as gaze guidance or increased deceleration. These are based on the complexity of the traffic situation, the current secondary task of the driver, and the gazes on the road [8]. Still, this approach does not simulate and predict the cognitive processes that are undergone during a takeover in different situations.

Other authors [13] proposed a driver model that focuses on the multitasking nature of the driving task. Within their model, general driving in different situations is modeled. When it comes to highly automated driving, the execution of different tasks in parallel is relevant to consider. However, their approach only models the driving task per se, but does not include the takeover in highly automated driving.

The same applies for other approaches [14,15] that implement cognitive models for the driving task. The cognitive model by [15] provides a useful approach to implement the general driving task within a cognitive architecture. He implemented the driving task based on several cognitive processes that are applied during the driving task to perceive the surrounding and perform motoric actions. However, this model does not include the takeover process, the whole dynamic driving environment, and the built-up of situation awareness.

An ACT-R (Adaptive Control of Thought-Rational; [16]) cognitive model for the takeover in highly automated driving [17] shows that such a model is able to represent cognitive processes during a takeover. Different complexities of situations can already be handled very well by that model, but no spectrum of different behaviors is included. This is addressed within this paper.

The aim of the present paper is to show how a spectrum of driving decisions can be integrated into such a modeling approach. Depending on how the dynamic situation

unfolds and what kind of action and attention decisions are made. These two aspects will influence the build-up of the individual situation awareness. The model follows such a situated cognition approach and tries to capture the variety of driving decisions in takeover situations that would follow under reasonable considerations. The representation of such differences can be modeled using ACT-R. Some authors [18], for example, developed a cognitive model to simulate automated driving using reinforcement learning for adaptions on automated vehicle operations. This adaptation process is situated and very relevant for automated driving. However, related to takeover maneuvers, complex decision strategies and the build up of situation awareness are not the focus of that work. Furthermore, another study [19] followed a situated cognition account on how a situation unfolds and what kind of behavior would be reasonable regarding task and context. When behavior deviates from model expectations, further explanations (e.g., via neurophysiological information) are explored by the model. Another approach [19] uses a form of model-tracing to represent individual situated pilots' behavior. Nevertheless, this work is more focused on auditory attention than on visual.

Such complex cognitive driver models that incorporate driving differences need to be integrated into embedded systems [9] to support a safe and comfortable takeover. It is a method for higher-level assessment of the situation at hand that can be described as situation awareness [12].

### 1.2. Situation Awareness

Situation awareness as defined by [20] includes three main stages to gain an understanding of a situation. In the first stage, the perception phase, objects in the surrounding environment are perceived. The perception phase is especially important for the takeover as it includes the visual perception of relevant vehicles in the surrounding traffic environment. This is followed by forming an understanding of the current situation in the comprehension phase [20]. In this paper, the phase is understood as the process when information about the situation and the task are integrated to gain a deeper understanding of the whole situation. After this, it is important that the driver is able to project the future situation with the altered state of other vehicles in the surrounding traffic (depending on speed, lane change indications, etc.), called the projection phase [20].

Overall, situation awareness can be understood as interrelated perceptual decisions about the world [12]. These are cognitively integrated to base decisions on.

### 1.3. SEEV Theory

Based on the SEEV theory, visual attention distribution in workplace scenarios can be predicted [21] and whether unexpected stimuli are perceived [22]. One approach [23] assessed the plausibility of SEEV in a driving scenario. SEEV (Salience Effort Expectancy Value) uses four parameters to describe attention-attracting properties. These are a combination of the bottom-up factors, salience and effort, and the top-down factors, expectancy and value. The workplace is subdivided into areas of interest (AOI). Value corresponds to the task relevance of an AOI and expectancy to how often this information changes. This indicates how often an AOI needs to be sampled in order to get the relevant information. For a driver, the view through the front window is, for example, a highly relevant area of interest. Nevertheless, a passenger doing a non-driving-related task (NDRT) will find his tablet display, for example, more important. Thus, the passenger will less frequently look through the front window (low expectancy, low value). However, this would still be more often than through the rear window, as it is closer to the current point of fixation, needing less effort to look there.

By understanding driver cognition and the state of situation awareness during the takeover, cooperative systems in highly automated driving can be developed. Such cooperative systems adjust both-sided (driver–system and system–driver) and can support the human driver directly and on an individualized level. A cognitive model that integrates situation awareness and SEEV enables a better prediction of cognitive processes. Especially to predict the process of building up situation awareness after a takeover request, the fusion of cognitive modeling and the SEEV theory is highly important. It allows the usage of AOIs in combination with human cognitive processes. Based on such a cognitive model, a better approximation to vehicle system dynamics can be developed and the cooperation between vehicle and driver enhanced. The goal in this paper is to show that the present cognitive model is able to (a) represent cognitive processes during a takeover, (b) interact with dynamic traffic environments, (c) use AOIs for visual perception based on the SEEV theory, (d) predict deviations that represent a spectrum of different behaviors in dependence of the current situation, (e) represent situation awareness during a takeover, and (f) match empirical data.

## 2. Materials and Methods

To simulate human cognition and represent corresponding cognitive processes, ACT-R [24] is used. The architecture ACT-R explains how multiple modules that represent cognitive processing units such as memory and auditory/visual perception are integrated to produce behavior in complex tasks [16]. Models within such an architecture can generate novel predictions and helps explain human behavior [25]. To represent the takeover in driving, the standard ACT-R modules are used. The procedural system is used in this work for the takeover process, as productions enable the most adaptive response from any state of the modules [26]. The original ACT-R visual module is replaced by a new vision module called SEEV-VM (Salience Effort Expectancy Value-Vision Module [27]). In contrast to the standard module that identifies objects in the visual field, the SEEV-VM uses AOIs that more closely represent human visual attention. The following sections describe the model environment, the new SEEV-VM module, and how the cognitive model is built up.

### 2.1. Graphical User Interface

The cognitive model interacts with a graphical user interface (GUI). In this GUI, different traffic scenarios are represented. Two visualizations exist, the in-car view (Figure 1) and the bird's-eye view (Figure 2).

The in-car view (Figure 1) shows the simulation and the AOIs that the model interacts with. This view is synchronized (y-position dependency) to the bird's-eye view [28]. The bird's-eye view is for visualization and control of the surrounding traffic by the modeler (Figure 2). Figure 2 shows the setup of the surrounding traffic environment when the takeover is requested. The vehicles in the GUI move different pixels per time-frame, resembling the different speed that the traffic environment in the empirical study has. The traffic environment resembles a six-lane highway (three lanes per direction), with the ego-vehicle driving on the center lane. The vehicles in the GUI move several pixels per time-frame, resembling the different speed that the traffic environment in the empirical study has.

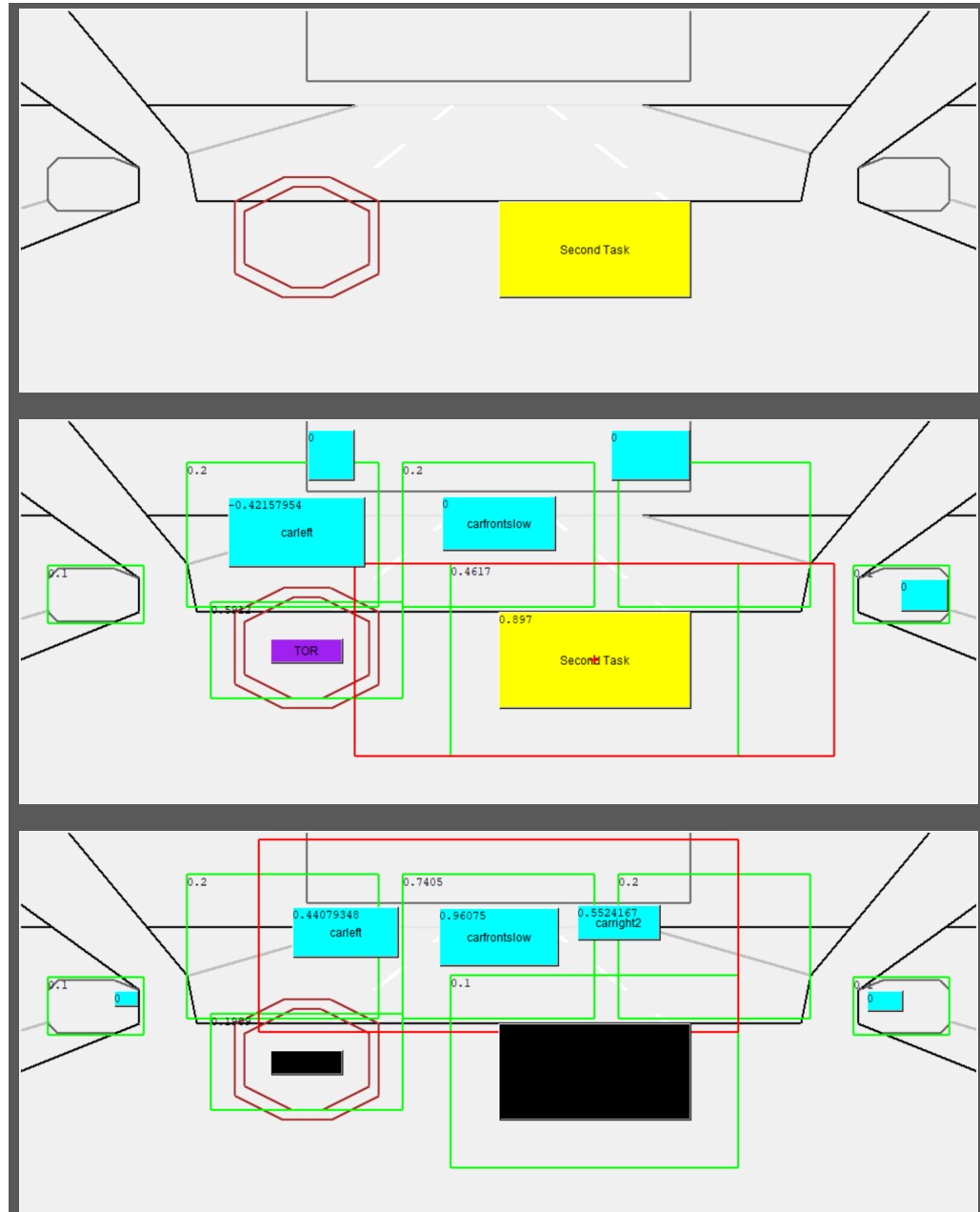

**Figure 1.** Original screenshot of the graphical user interface that the cognitive model interacts with. The screenshot represents the areas of interest (green frames), the field of view (red frames), and vehicles in the traffic environment (solid blue/green squares). Depicted example scenes show from top: the lane graphics and the NDRT, a complex traffic scenario with TOR (takeover request; violet solid square), and NDRT active (yellow solid square), traffic scenario with NDRT and TOR deactivated (black solid squares; source: own figure).

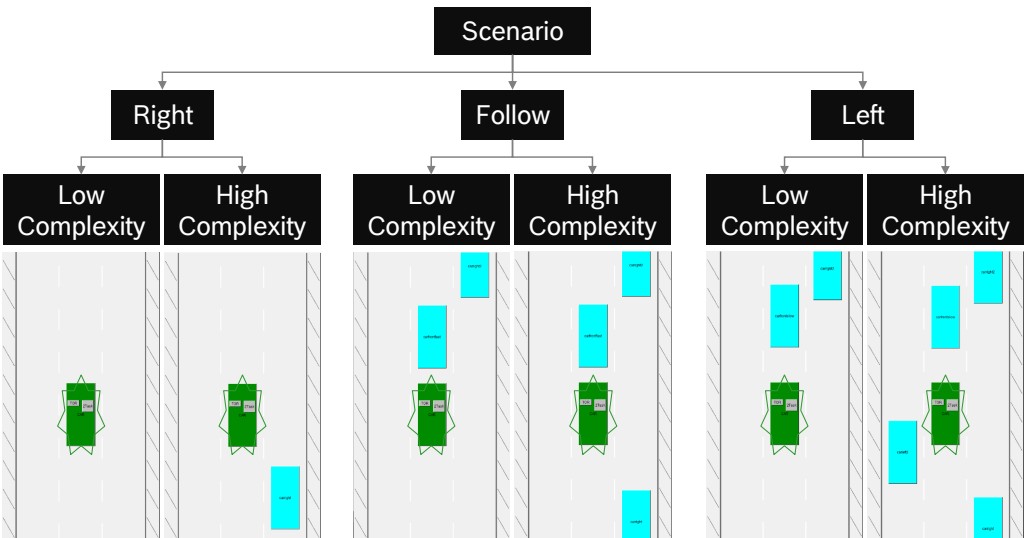

**Figure 2.** Representation of traffic scenarios in the graphical user interface representing the ego-vehicle (star-green) and relevant vehicles in the surrounding traffic environment at the takeover point (blue/green; source: own figure).

### 2.2. Scenarios

Every scenario starts with an NDRT until the takeover request, located behind the steering wheel, appears. Traffic is set up to make three different maneuvers necessary (lane-change left, right, and car following).

Each maneuver is set up in a high or low complex surrounding traffic environment. This results in overall six different scenarios. Figure 2 gives an overview of the scenarios and the corresponding traffic in the bird's-eye view. In order to trigger the corresponding maneuver, the speed of the vehicles differs between the different scenarios. In the following condition, for example, the leading vehicle has a higher speed and is thus followed, whereas the in the left condition with low complexity, the leading vehicle is significantly slower and triggers a lane change to the left.

### 2.3. Cognitive Model

The cognitive model represents a spectrum of cognitive processes during a takeover in dynamic traffic environments that builds up a representation of situation awareness. The model captures processes upon the situation as it unfolds and includes decision-points where several options are possible and chosen by chance. The model uses the ACT-R default parameters in all cases. No additional parameter fitting is done. The default production specified action time is 50 ms. Furthermore, at this stage of the model, learning is not integrated.

Within the model, a bandwidth of different driving strategies is implemented that the model applies during a takeover task. Some drivers, for example, look over the shoulder before changing a lane. Others prefer using the mirrors, others do both. Furthermore, drivers differ in their caution when taking over the driving task. Some drivers, for example, are more cautious (more braking) than others (acceleration). Based on the surrounding traffic, the model perceives the situation and makes corresponding action decisions. For the perception of the blind spot, a motoric movement is used, indicating the view over the shoulder. No AOIs are set for the blind spot as it is a yes/no perception without visual tracking. The model represents different decision points where several options are eligible with the same probability. This results in a divergent rather than a deterministic model that produces a high spectrum of behavioral representations. This kind of model includes all kinds of reasonable decisions considering the current situation and the information that has already been collected. This model could at a later point be used for model

tracing. In the model tracing approach (e.g., [29]), actions of the user are traced and the underlying model provides means to figure out the appropriate productions that lead to the actions. Thus, gaze and actions drive the tracing process and provide hypothesis about the current cognitive state and representation that explain the behavior. With this information, appropriate assistance or intervention can be given whenever necessary. At this point, the model is required to show a variety of behaviors that is in accordance with the situation and the sequence of actions to see whether this shows reasonable performance compared to participants' performance. In addition, the model builds up situation awareness that is represented in a chunk. The current state of situation awareness influences decisions at decision points where premises apply based on the dynamics of the traffic situation.

The following sections describe the visual guidance of the scene perception, the representation of situation awareness within the model, and the corresponding decision flow.

### 2.4. Visual Guidance

Visual perception is an integral part of the cognitive model to build up situation awareness. Three requirements to model visual guidance are identified for this model. First, the detection of unexpected, not explicitly requested visual features such as alarm signals. Second, prediction-based gaze control, and third, the realistic simulation of eye movements in visually rich and highly dynamic scenes.

In this approach, the SEEV vision module (SEEV-VM; [27]) is used as a replacement for the default ACT-R vision module. The following properties of the original SEEV theory are implemented in this vision module. Top-down gaze control in the form of areas of interest (AOI). These are defined by the modeler. However, the original SEEV theory does not provide an approach to predict visual guidance in dynamic scenes, as realized here. Visual guidance here is composed of salience, effort, and relevance. The equation below shows how to compute visual guidance.

$$guidance\ value\ =\ relevance + salience - effort \tag{1}$$

In the following section, salience, relevance and effort are explained in detail.

Salience: Visual salience in general is considered as the physical attractiveness of an object. It is based on the contrast of an object to the close surrounding. E.g., a flashing red light in an otherwise indistinguishable gray space is highly salient. It can be calculated, but this would require a very extensive simulation of the environment.

Relevance: Every AOI has a relevance that is a combination of value and expectancy (which are the factors in the original account). The value reflects the task-related importance of expected information at the corresponding location. Expectancy describes here the amount of relevant objects and the frequency of change within an AOI. In the original SEEV theory, an AOI contains information of every object within this AOI. In the SEEV-VM implementation, objects lie within AOIs and are attended and encoded individually. Therefore, relevance changes dynamically. If an object within an AOI is attended, the relevance of this AOI decreases by the consumption value. Accordingly relevance increases over time by the refresh rate. Both values correspond to the expectancy value of the original SEEV theory and have to be set by the modeler.

Effort: Saccadic effort is calculated based on the spatial distance of an target object to the current point of fixation. Effort is high when objects are far away.

Within the GUI, scenarios are composed of objects that are defined by the modeler. The modeler describes objects by assigning salience values and contextual meaning (e.g., car, lane marking) to them. The SEEV-VM approach frees the modeler of designing the visual perception processes in detail but uses an indirect mechanism based on the guidance value Formula (1).

By using SEEV-VM, the model occasionally checks the vehicle interface while primarily conducting an NDRT. This is realized by two AOIs. One AOI for the NDRT display and one for the takeover request display (Figure 1). Once the takeover request signal is activated, it has a high salience value. This increases the guidance value of the takeover request signal.

In this way, a bottom-up process can trigger a top-down process that initiates a task switch. Following this task switch, relevance values of AOIs are readjusted.

### 2.5. Situation Awareness Representation

The process to build up situation awareness within this model is a top-down process. In contrast to [15], this model includes a more complex approach to build up situation awareness that is more goal-directed rather than general maintenance. Furthermore, the takeover in highly automated driving requires to reconstitute situation awareness. It is dependent on the current situation and is built-up progressively. The model starts after the takeover request is triggered. Using the SEEV-VM, the AOI of the front lane has the highest relevance in the beginning. The first goal, after the takeover request has been attended, is to build up situation awareness. After the front lane is perceived, the front part of situation awareness is built-up and the information stored in a mental representation is called a chunk. This chunk is located in the imaginal buffer. It is able to store information (in the slots) about the position of vehicles on the three lanes, representing current situation awareness. Based on the information that is stored in this chunk, decisions are made. Depending on the current state of situation awareness, different AOIs rise in relevance, depending on the current goal. Thus, if there is no upcoming vehicle, for example, the right lane becomes an AOI with a high relevance to check whether it is free (obligation to drive on the right). Based on the information about the situation of the perceived AOI, situation awareness is updated and the goal adapted. The information that the model perceives in the dynamic situations is matched to conditions of different production rules.

### 2.6. Decision Flow

In order to provide an understanding of how such a production rule looks like, Figure 3 provides an exemplary production rule. Only productions that fit to the information in the buffers (these are specified in the upper half) can fire and changes, e.g., the goal, the imaginal buffer or requests to the visual-organization (control concept of the SEEV-VM) are conducted.

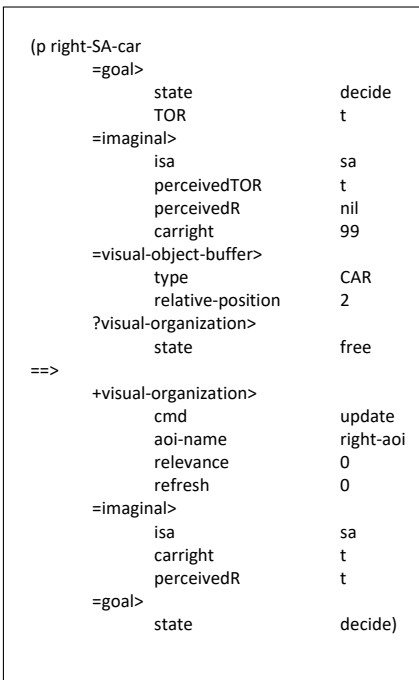

**Figure 3.** Exemplary production rule that fires when a car on the right oncoming lane is perceived while situation awareness is build up (99 = dummy variable; t = true; TOR = takeover request; R = right; aoi = area of interest; sa = situation awareness; source: own figure).

The production fires if the current goal is to make a decision and a takeover request has been perceived. This information is stored in the imaginal buffer together with the information that the right lane has not been perceived yet (nil). Therefore, the value for a vehicle on the right is still the dummy (99). This indicates that no car on the right has been perceived yet, neither has the right been attended. Furthermore, a visual object has been perceived of the type CAR on the right oncoming lane (Position 2) and the visual organization has to be free. If all this is the case, then (==>) the right AOI's relevance is set to 0, as it has just been attended. Furthermore, the imaginal buffer stores the information that the right lane has been perceived and that there is a car driving on the front oncoming lane (t). The goal is still to make an action decision. In case the conditions match a certain production, as described above, this production will be carried out. As in this version of the model no utilities are used, one of several possible productions is chosen by chance. Figure 4 represents the whole decision flow. First, the model sets the pre-defined relevant AOIs for NDRT, HMI, the three lanes (left, oncoming/front, right), and left and right mirror. As the model represents the takeover, it starts with an NDRT (Figure 4). The NDRT of the model equals the automated mode when the driver engages into an NDRT. The model initially attends the NDRT that is active (*NDRT*).

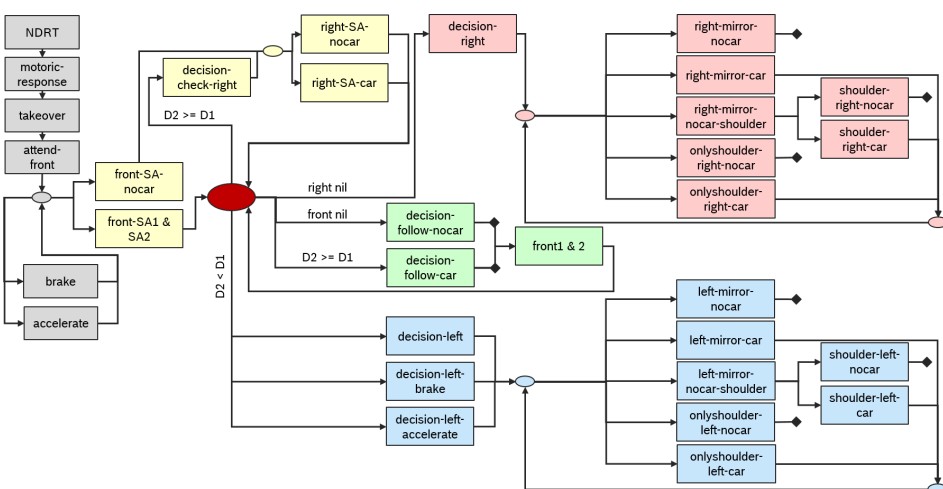

**Figure 4.** Structure of the Cognitive Model. Boxes with productions that represent the procedural system. Arrows show possible steps in the process. Ovals mark process points, where the goal in the goal buffer is changed. Colors mark different steps in the process to build up situation awareness (SA; gray = takeover and first motoric reactions, yellow = building of first SA to base a decision on, blue = lane change right, green = following, red = lane change left; D = distance; source: own figure).

As soon as the takeover request appears, the model responds with first automated reactions (gray boxes; Figure 4). The takeover request is recognized, the model simulates moving the hands to the steering wheel (*motoric-response*), and the model takes over the driving task (*takeover*). The front is attended (*attend-front*) and another motoric reaction is executed. This motoric longitudinal driving reaction can either be an acceleration (*accelerate*) or a deceleration (*brake*). As the ACT-R architecture does not provide movements of the feet, the default ACT-R movement of the hands is used as substitute for a motoric action. Next, the situation awareness (SA) of the current traffic situation has to be built up (yellow boxes; Figure 4). If no upcoming vehicle is present (*front-SA-nocar*), the right side is attended and it is decided whether there is a car (*right-SA-car*) driving on the right lane or not (*right-SA-nocar*). In cases, where an upcoming vehicle is present, this vehicle is attended twice. The two distance points (D; Figure 4) are contrasted. Based on this information, the model goes back into the central decision stage (red oval; Figure 4) where it takes an action decision based on the collected information. If the distance increases or stays the same, the right side is attended as described above. If the distance decreases, the model decides to

overtake the leading vehicle with a lane-change to the left. Thus, it starts checking the left lane for overtaking vehicles.

2.6.1. Maneuver Decision Lane-Change to the Right

Based on the obligation to drive on the right, a decision to make a lane change to the right (blue boxes; Figure 4) is made when the right lane is free (*decision-right*). This is the case when no car is detected on the right lane and right equals nil. In case a car on the right is detected, right would not equal nil and a different decision is made. As soon as the lane-change right decision is made, the lane-change maneuver is prepared. Three different behavioral options to make sure no car is driving in the right back are possible. The first option is to check only the mirror. If no vehicle is detected in the right mirror (*right-mirror-nocar*), the lane change is executed. In case there is a vehicle in the mirror (*right-mirror-car*), a new decision between the three behavioral options is made. The second optional behavior is to check the mirror and if there is no car, look over the shoulder before executing the lane change (*right-mirror-nocar-shoulder*). When there is still no car detected with the shoulder check (*shoulder-right-nocar*), the lane change is executed. If a car is recognized due to the over-shoulder check (*shoulder-right-car*), a new behavioral decision for the right back area is made. The third behavioral option is to look only over the shoulder. Again, if no car is in the field of view, the lane change is executed (*onlyshoulder-right-nocar*) and if there is a car detected, a new behavioral decision is made (*onlyshoulder-right-car*).

2.6.2. Maneuver Decision Car Following

Figure 4 represents car-following productions with green boxes. If no upcoming vehicle is present but the right lane occupied (*right-SA-car*), the decision to stay on the current lane is made (*decision-follow-nocar*). If an upcoming vehicle is present, the car in front is attended twice and distances are compared (*front-SA1 & SA2*). In case the distance (D; Figure 4) to the car in front increases or stays the same, the right lane is checked for a vehicle (*decision-check-right*). The above-described procedure for a lane change to the right is performed, if no car is detected on the right (*right-SA-nocar*). In case there is a car present on the right lane (*right-SA-car*) and the distance (D; Figure 4) to the car in front increases or stays the same, car following is decided (*decision-follow-car*). The front is attended again (*front1 & 2*) to update the status of the vehicle in front.

2.6.3. Maneuver Decision Lane-Change to the Left

In case the leading vehicle is driving slower than the ego-vehicle, a decision to overtake is made (red boxes; Figure 4). Three options are possible when deciding to do a lane change to the left. The lane change can be executed without changing the current speed (*decision-left*), slower (*decision-left-brake*) or faster (*decision-left-accelerate*). All three options are followed by attending the left back. As in the condition for the right maneuver, three behavioral patterns are possible. Those are: first, checking the mirror only (*left-mirror-car & left-mirror-nocar*). Second, checking the mirror (*left-mirror-nocar-shoulder*) and if there is no car, rechecking the blind spot with the over-shoulder view (*shoulder-left-car & shoulder-left-nocar*). Third, using the over-shoulder view only (*onlyshoulder-left-car & onlyshoulder-left-nocar*). There can either be a vehicle in the attended area (*...-car*) or not (*...-nocar*). If there is no vehicle detected, the lane change is executed. If a vehicle is detected in the relevant area, a new behavioral decision to attend the left back is made.

*2.7. Empirical Data*

In order to validate predictions of the cognitive model, these have been compared to a driving simulator study in Figure 5. As driving simulators are identified as the prevalent testing environment in the field of usability assessments of HMIs and an efficient and risk-free alternative to real driving experiments [30], this method was chosen to validate the cognitive model. The study was conducted in spring 2019 and the test design, methods,and technical functionality were tested in a pre-study. The study was conducted

with N = 20 participants (13 male, 7 female) with a mean age of 26.2 years (SD = 2.69). Most participants drive on average 30 min per day, mostly on highways and indicate a moderate driving style [31,32]. Each participant ran through every scenario three times. Thus, empirical data represent N = 60 samples per scenario. Commonly, it is proposed to run a model 100 times [33] with N = 20 participants. However, based on the different scenarios, the model is run through each scenario 60 times to have comparable sample sizes. The implementation of the study was approved by the ethics committee of the TU Berlin in April 2019 and Robert Bosch GmbH.

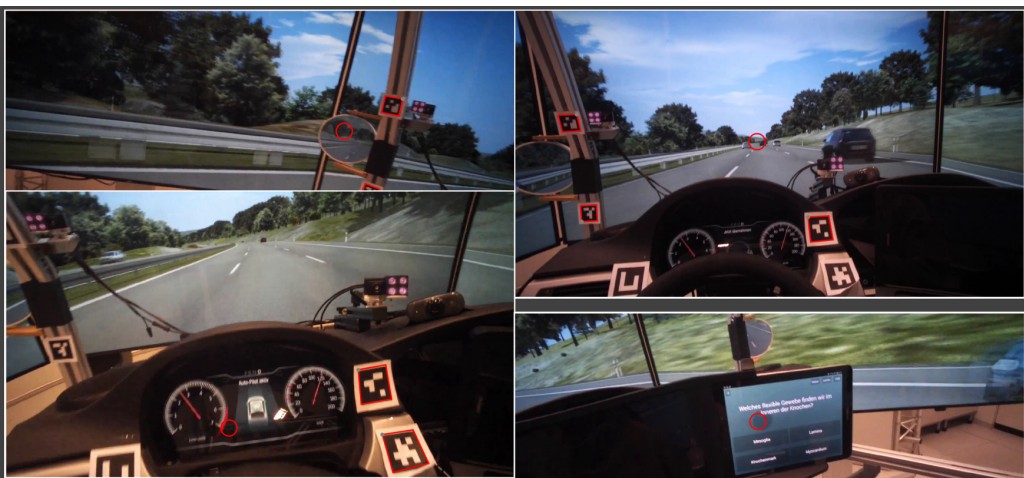

**Figure 5.** Depiction of the driving simulator study. Source: [31].

## 3. Results

The model is able to interact with different dynamic environments and make the most common maneuver decisions (lane change left, right, and following; [34]) as shown in Figure 4. The specific results of these maneuver decisions are described in the sections below.

The model is run through each scenario 60 times (N = 60) with parameters set to initial values each time. The vehicles in the GUI move different amounts of pixels per time-frame. This resembles the different speeds within the traffic environment of the empirical study in which the ego-vehicle drove at a speed of 120 km/h when the takeover was triggered. Results show that the model is able to make maneuver decisions in different scenarios. Prediction times show variations depending on the maneuver and the complexity of the surrounding traffic environment (Figure 6). The rule firing times vary due to a different selection of production rules depending on the driver and the traffic environment. Figure 6 represents these differences in execution times by indicating the points NDRT, takeover, decision, and execution of an action. The buildup of situation awareness mainly happens between the NDRT and the takeover, but it is an ongoing process and essential to make and execute an action in a complex environment.

In the following, results of the three different scenarios are represented in tables in separate sections. The column *Events* of the tables (Tables 1–3) indicates how often a production is fired. Productions can be fired more than once in a case. Braking, for example, can fired more often, indicating a stronger deceleration. The tables start with the listing of the first action. Before this, the takeover request appears and is perceived. In the end, model predictions are compared to empirical data. Reaction times that are compared to each other always correspond to the initiation of an action on the timeline of the takeover. Based on these action initiations, cognitive model predictions and empirical data are compared.

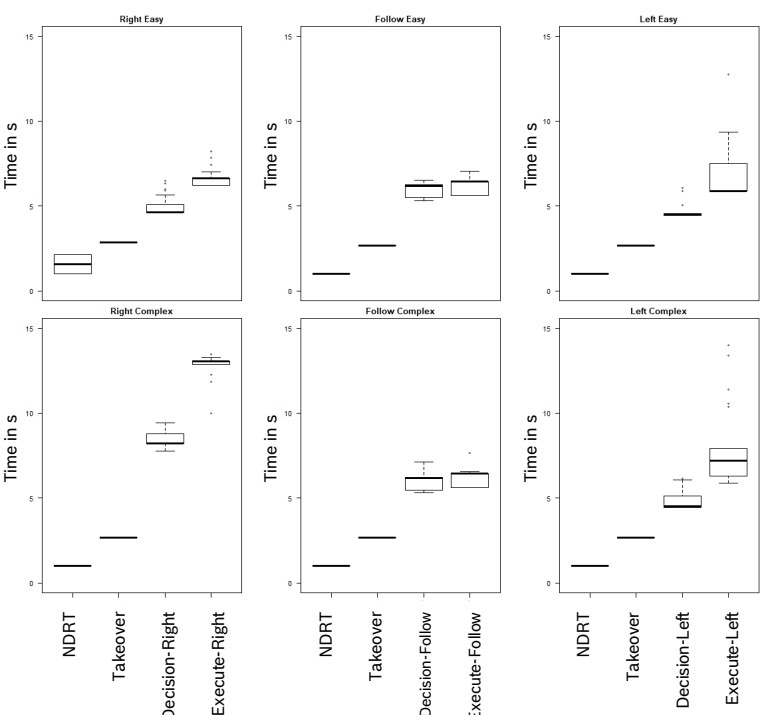

**Figure 6.** Box-whisker plot of model predictions for each task in each condition. The y-axis shows the timeline in seconds (s) including the NDRT, the takeover, the corresponding action decision, and the action execution (source: own figure).

**Table 1.** Predicted time statistics of the cognitive model with a lane-change right maneuver. Representation of minimum (Min), mean, and maximum (Max) values over *N* = 60 trials. Wrong decisions are in lower case letters (source: own table).

| Production | Events | Predicted Time | | | |
|---|---|---|---|---|---|
| | | **Min** | **Mean** | **Max** | **SD** |
| *Right Low Complexity* | | | | | |
| ACCELERATE | 74 | 3.07 s | 3.49 s | 4.86 s | 0.51 |
| BRAKE | 77 | 3.07 s | 3.4 s | 34.86 s | 0.37 |
| FRONT-SA-NOCARFRONT | 60 | 3.36 s | 3.62 s | 4.98 s | 0.44 |
| RIGHT-SA-NOCAR | 60 | 4.53 s | 4.83 s | 6.43 s | 0.49 |
| DECISION-RIGHT | 60 | 4.58 s | 4.92 s | 6.48 s | 0.61 |
| RIGHT-MIRROR-NOCAR | 19 | 6.16 s | 6.38 s | 6.97 s | 0.31 |
| RIGHT-MIRROR-NOCAR-SHOULDER | 19 | 6.16 s | 6.52 s | 7.78 s | 0.55 |
| ONLYSHOULDER-RIGHT-NOCAR | 22 | 6.16 s | 6.46 s | 7.78 s | 0.44 |
| SHOULDER-RIGHT-NOCAR | 19 | 6.57 s | 6.92 s | 8.19 s | 0.56 |
| EXECUTE-RIGHT | 60 | 6.21 s | 6.63 s | 8.24 s | 0.5 |
| *Right High Complexity* | | | | | |
| ACCELERATE | 154 | 2.87 s | 3.64 s | 7.75 s | 0.88 |
| BRAKE | 146 | 2.87 s | 3.58 s | 6.98 s | 0.78 |
| FRONT-SA-NOCARFRONT | 1 | 7.8 s | 7.8 s | 7.8 s | - |
| FRONT-SA-NOCARFRONT-PERCEIVED | 59 | 3.42 s | 3.92 s | 6.27 s | 0.62 |
| RIGHT-SA-NOCAR | 46 | 7.73 s | 8.41 s | 9.39 s | 0.43 |
| DECISION-RIGHT | 46 | 7.78 s | 8.49 s | 9.44 s | 0.44 |
| RIGHT-MIRROR-NOCAR | 16 | 9.95 s | 12.68 s | 13.02 s | 0.75 |
| RIGHT-MIRROR-NOCAR-SHOULDER | 16 | 11.38 s | 12.87 s | 13.23 s | 0.43 |
| ONLYSHOULDER-RIGHT-NOCAR | 14 | 9.95 s | 12.73 s | 13.23 s | 0.82 |
| SHOULDER-RIGHT-NOCAR | 16 | 11.79 s | 13.1 s | 13.43 s | 0.39 |
| EXECUTE-RIGHT | 46 | 10 s | 12.89 s | 13.48 s | 0.69 |
| decision-follow-nocar | 14 | 10.66 s | 11.55 s | 13.02 s | 0.71 |
| execute-follow | 14 | 10.71 s | 11.6 s | 13.06 s | 0.71 |

Right Maneuver Scenarios: The first scenario that the model interacts with is the condition in which a lane change to the right is the best maneuver decision. Table 1 shows that in the complex condition, a maneuver decision is made 3.57 s (42%) later and the execution 6.26 s (48.6%) later. In addition, the model accelerates more often in the complex condition where a vehicle is present in the right back of the ego-vehicle. In the complex scenario, the model makes the wrong action decision to stay on the center lane in 14 out of 60 cases although the right oncoming lane is free.

Follow Maneuver Scenarios: In scenarios in which the best maneuver decision is car following, the model does not show significant differences between complexities. Model predictions do only show minimal difference in the time until an action decision is made ($M_{D.FLow} = 5.94$ s; $M_{D.FHigh} = 5.95$ s; Table 2). In both situations, only the leading and the right vehicles are attended. This result shows that only if vehicles in the surrounding traffic environment are relevant, they have an impact on the cognitive processes of the driver. In both car-following cases, the model makes a wrong decision. This is a lane change to the right ($M_{D.Wrong} = 5.67/5.81$ s; Table 2) although the right lane is occupied.

**Table 2.** Predicted time statistics of the cognitive model with a follow maneuver. Representation of minimum (Min), mean, and maximum (Max) values over $N = 60$ trials. Wrong decisions are in lower case letters (source: own table).

| Production | Events | Predicted Time | | | |
|---|---|---|---|---|---|
| | | Min | Mean | Max | SD |
| Follow Low Complexity | | | | | |
| ACCELERATE | 84 | 2.87 s | 3.11 s | 3.37 s | 0.22 |
| BRAKE | 96 | 2.87 s | 3.12 s | 3.37 s | 0.2 |
| FRONT-SA1 | 60 | 3.42 s | 3.44 s | 3.47 s | 0.03 |
| FRONT-SA2 | 60 | 4.81 s | 5.28 s | 5.62 s | 0.4 |
| DECISION-CHECK-RIGHT | 376 | 4.86 s | 5.55 s | 6.37 s | 0.42 |
| RIGHT-SA-CAR | 59 | 5.26 s | 5.84 s | 6.42 s | 0.42 |
| DECISION-FOLLOW-CAR | 59 | 5.31 s | 5.94 s | 6.52 s | 0.43 |
| FRONT1 | 59 | 5.62 s | 6.13 s | 7.04 s | 0.42 |
| decision-right | 1 | 5.76 s | 5.76 s | 5.76 s | - |
| execute-right | 1 | 6.49 s | 6.49 s | 6.49 s | - |
| Follow High Complexity | | | | | |
| ACCELERATE | 87 | 2.87 s | 3.13 s | 3.37 s | 0.2 |
| BRAKE | 93 | 2.87 s | 3.01 s | 3.37 s | 0.21 |
| FRONT-SA1 | 60 | 3.42 s | 3.44 s | 3.47 s | 0.02 |
| FRONT-SA2 | 60 | 4.82 s | 5.3 s | 5.62 s | 0.4 |
| DECISION-CHECK-RIGHT | 388 | 4.86 s | 5.57 s | 7.02s | 0.45 |
| RIGHT-SA-CAR | 59 | 5.26 s | 5.86 s | 67.07 s | 0.43 |
| DECISION-FOLLOW-CAR | 59 | 5.31 s | 5.95 s | 7.12 s | 0.43 |
| FRONT1 | 59 | 5.62 s | 6.16 s | 7.65 s | 0.44 |
| decision-right | 1 | 5.81 s | 5.81 s | 5.81 s | - |
| execute-right | 1 | 7.51 s | 7.51 s | 7.51 s | - |

Left Maneuver Scenarios: The third maneuver option is a lane change to the left. Results show that in both scenarios, the action decision is taken at almost the same time ($M_{D.LLow} = 4.79$ s; $M_{D.LHigh} = 4.77$ s). This is due to the fact that the decision is based on the slower leading vehicle. For the action execution on the other hand, the vehicles on the left lane are relevant. Hence, 0.47 s (6%) more are needed to execute the action in the complex scenario (Table 3). In the easy scenario, the model makes 14 decisions for a lane change to the right. In the complex scenario, 12 times a lane change to the right is executed. This decision is suboptimal in the given traffic situation and thus considered as a wrong decision ($M_{D.Wrong} \approx 3.7$ s; Table 3).

**Table 3.** Predicted time statistics of the cognitive model with a lane change left maneuver. Representation of minimum (Min), mean, and maximum (Max) values over $N = 60$ trials. Wrong decisions are in lower case letters (source: own table).

| Production | Events | Predicted Time | | | |
|---|---|---|---|---|---|
| | | **Min** | **Mean** | **Max** | **SD** |
| *Left Low Complexity* | | | | | |
| ACCELERATE | 105 | 2.87 s | 3.18 s | 3.65 s | 0.23 |
| BRAKE | 93 | 2.87 s | 3.15 s | 3.65 s | 0.26 |
| FRONT-SA1 | 46 | 3.6 s | 3.64 s | 3.7 s | 0.05 |
| FRONT-SA2 | 46 | 4.41 s | 4.83 s | 5.83 s | 0.64 |
| DECISION-LEFT | 14 | 4.46 s | 4.74 s | 6.08 s | 0.55 |
| DECISION-LEFT-ACCELERATE | 12 | 4.46 s | 4.96 s | 5.98 s | 0.7 |
| DECISION-LEFT-BRAKE | 20 | 4.46 s | 5 s | 5.98 s | 0.69 |
| LEFT-MIRROR-NOCAR | 17 | 5.84 s | 6.73 s | 9.31 s | 1.37 |
| LEFT-MIRROR-NOCAR-SHOULDER | 12 | 5.84 s | 7.1 s | 9.31 s | 1.49 |
| ONLYSHOULDER-LEFT-NOCAR | 17 | 5.84 s | 6.34 s | 9.31 s | 1.18 |
| SHOULDER-LEFT-NOCAR | 12 | 6.66 s | 8.88 s | 12.7 s | 2.38 |
| EXECUTE-LEFT | 46 | 5.89 s | 7.2 s | 12.75 s | 1.91 |
| decision right | 14 | 3.7 s | 3.71 s | 3.85 s | 0.04 |
| execute right | 14 | 5.07 s | 5.14 s | 5.28 s | 0.1 |
| *Left High Complexity* | | | | | |
| ACCELERATE | 94 | 2.87 s | 3.15 s | 3.65 s | 0.24 |
| BRAKE | 103 | 2.87 s | 3.17 s | 3.65 s | 0.25 |
| FRONT-SA1 | 48 | 3.6 s | 3.64 s | 3.7 s | 0.04 |
| FRONT-SA2 | 48 | 4.41 s | 4.79 s | 6.03 s | 0.61 |
| DECISION-LEFT | 16 | 4.46 s | 4.86 s | 6.13 s | 0.66 |
| DECISION-LEFT-ACCELERATE | 15 | 4.46 s | 4.72 s | 6.13 s | 0.47 |
| DECISION-LEFT-BRAKE | 17 | 4.46 s | 5 s | 6.18 s | 0.68 |
| LEFT-MIRROR-CAR | 21 | 5.22 s | 5.65 s | 8.48 s | 0.98 |
| ONLYSHOULDER-LEFT-CAR | 23 | 5.22 s | 5.65 s | 8.48 s | 0.94 |
| LEFT-MIRROR-NOCAR | 23 | 5.84 s | 6.95 s | 13.36 s | 1.78 |
| LEFT-MIRROR-NOCAR-SHOULDER | 19 | 5.84 s | 7.6 s | 13.36 s | 2.37 |
| ONLYSHOULDER-LEFT-NOCAR | 9 | 5.84 s | 7.35 s | 13.36 s | 2.38 |
| SHOULDER-LEFT-NOCAR | 16 | 6.65 s | 8.74 s | 13.97 s | 2.36 |
| EXECUTE-LEFT | 48 | 5.89 s | 7.67 s | 14.02 s | 2.21 |
| decision-right | 12 | 3.7 s | 3.7 s | 3.8 s | 0.03 |
| execute right | 12 | 7.46 s | 7.63 s | 7.87 s | 0.21 |

*Comparison to Empirical Data*

Predictions of the model in the scenarios are compared to a simulator study with $N = 20$ participants [31]. Each participant ran through every scenario three times. Thus, empirical data represent $N = 60$ samples per scenario.

In the empirical simulator study, participants took a decision in the same scenarios as the model did. In this paper, empirical decision times until the first action decision is made (including errors; between subject) are used, rather than times until participants take the right action decision as in [31]. This is of high interest, as the model is able to predict errors. Results of the traffic simulator study and model predictions are shown in Table 4 and Figure 7. The comparison between predicted and empirical values below shows that model predictions are already very close to empirical data in most conditions, but need to be improved for right complex and left complex scenarios. Figure 7 shows model predictions and empirical data in the different conditions for the main takeover points. The gray (empirical) and white (model) boxes represent the standard deviations of the data that are additionally listed in Table 4. Mean values of model predictions are in general close to the mean values of the empirical data. However, the standard deviations in empirical data are much higher than the model data. In the right easy scenario, participants took between 0.73 and 12.42 s to take an action decision. The model, in contrast, represents driver differences that deviate between 4.58 and 6.48 s. Furthermore, empirical data show

higher deviations in the time to take over the driving task. At this stage of the takeover, model predictions do not show any variation. In the right complex and the follow scenarios, participants are faster than the model. In the other three conditions on the other hand, the model is faster. Overall, predictions of right easy and left complex scenarios are mostly different from empirical data. Here, the model is faster than participants. Especially in the left complex conditions, standard deviations of empirical data are very high (3.11). In contrast, model predictions do not show higher standard deviations in the left complex scenario than in the other conditions. The highest standard deviation that the model is able to represent is in the right complex scenario (1.4). In empirical data, the complex conditions show higher standard deviations than the easy conditions. This difference is also found in model predictions except for the left scenarios. Here, the model predicts less differences between drivers in the complex scenarios than in the easy ones.

**Table 4.** Action decision (including wrong decision) times of empirical data vs. model predictions. Representation of minimum (Min), mean, maximum (Max) values, standard deviation (SD) for right (R), follow (F), and left (L) scenarios in easy (E) and complex (C) conditions (source: scharfeTRF).

| Scenario | Empirical Data | | | | Model Predictions | | | |
|---|---|---|---|---|---|---|---|---|
| | Min | Mean | Max | SD | Min | Mean | Max | SD |
| R E | 0.73 s | 5.06 s | 12.42 s | 2.07 | 4.58 s | 4.92 s | 6.48 s | 0.5 |
| R C | 2.1 s | 5.34 s | 12.63 s | 2.23 | 7.78 s | 9.2 s | 13.02 s | 1.4 |
| F E | 2.82 s | 5.09 s | 9.7 s | 1.79 | 5.31 s | 5.94 s | 6.52 s | 0.42 |
| F C | 1.5 s | 5.11 s | 23.54 s | 3.13 | 5.31 s | 5.95 s | 7.12 s | 0.43 |
| L E | 2.73 s | 4.84 s | 10.25 s | 1.73 | 3.7 s | 4.63 s | 6.08 s | 0.76 |
| L C | 2.5 s | 5.69 s | 21.1 s | 3.11 | 3.7 s | 4.63 s | 6.18 s | 0.72 |

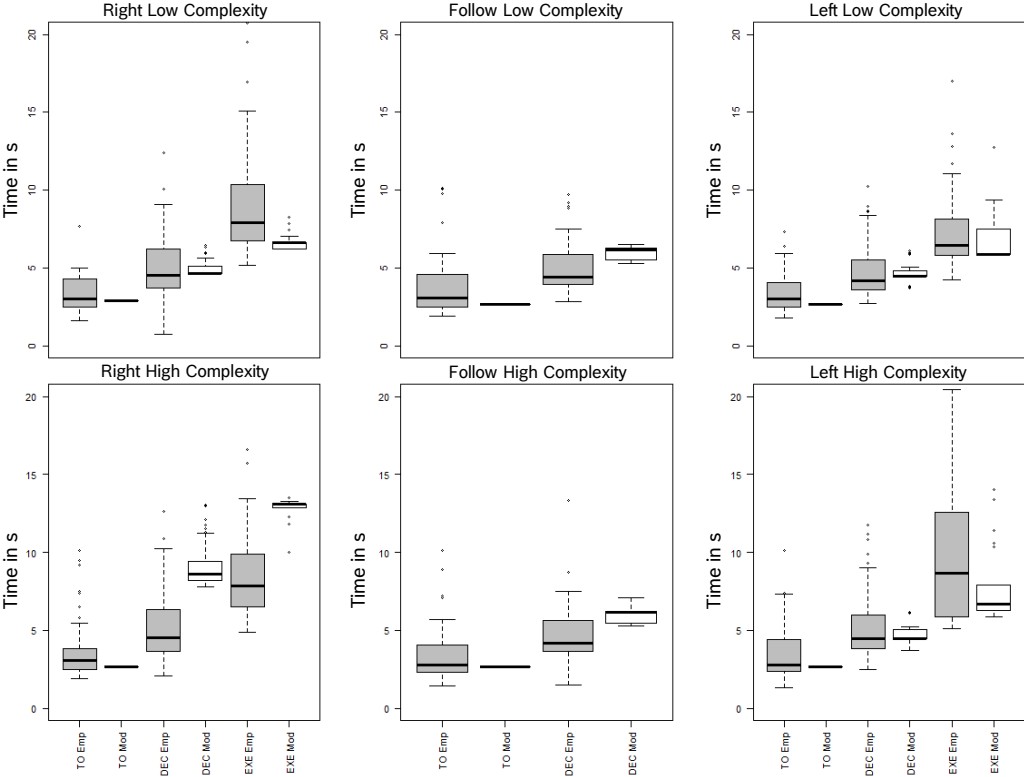

**Figure 7.** Box-Whisker plot of empirical data (Emp, gray) and model predictions (Mod, white) for the timeline in seconds (s) on the y-axis of the takeover (TO) and the time of the action decision (DEC). In right and left maneuver cases, the time of the action execution (EXE) is also included (source: own table).

Moreover, the model and participants make wrong decisions in the different scenarios. The model makes 42 wrong decisions in $N = 360$ (60 per scenario) takeovers. Participants of the empirical study made 13 wrong decisions in $N = 360$ (60 per scenario) takeover tasks. Wrong model decisions are mostly based on vehicles that have not been detected or too much acceleration/deceleration in the beginning. In the empirical study, wrong decisions are mostly based on overwhelmed drivers that decide to brake strongly (although the task is to keep the speed) and make a lane change to the right or follow the leading vehicle that is far slower than the ego-vehicle at the time of the takeover.

## 4. Discussion

The current model provides concrete operationalization of situation awareness, representing a plausible model of perceptual decisions. It is able to simulate cognitive processes during the takeover in highly automated driving from a non-driving-related task until the action execution. Interestingly, the model takes more time in the right complex maneuver scenario than in the left maneuver scenarios. As left maneuvers are usually more complex than a lane change to the right, lane changes to the left are assumed to take more time than to the right, due to more validation steps that are required to take over. Within the right complex maneuver, however, the vehicle in the right back had to be tracked and the speed calculated to make sure that it is safe to perform the lane change. As vehicles driving on the right are the slowest, the tracking of the vehicle in the right back takes longer until a safe distance to change the lane is reached. In the left maneuvers, the model only has to perceive that the right lane is occupied and the leading vehicle is slower. Overtaking vehicles in the left complex scenario also have to be perceived, but they overtake faster, and thus, the lane change can be executed.

Altogether, the model provides the basis to develop cognitive assistance systems for highly automated driving in the following way. As early warnings help distracted drivers to react more quickly [35], it is especially important to identify the takeover readiness and derive relevant adaptions. Those could be, for example, handing over step by step, visualizing the best maneuver option or indicating important aspects of the traffic environment to bring the driver back into-the-loop and enable a safe takeover. In combination with information of the vehicle sensors and eye tracking or physiological data of the driver, this cognitive model could directly interact with assistance systems by providing information about the current cognitive state of the driver.

Figure 8 shows how sensory data of the vehicle and wearables can be combined with the cognitive model to provide a situational adaption of assistance systems in highly automated driving. The integration of eye tracking into cognitive model predictions also allows the integration of the situation awareness status of the driver. Thus, a driver who is partly or completely in-the-loop would have already perceived certain objects of the traffic environment. These objects are stored in the declarative module. This represents the amount to which situation awareness is already created. This highlights the benefit of integrating a cognitive model into the development of adaptive assistance systems in highly automated driving. The model is able to represent a spectrum of different driving behaviors, resulting in deviating time that is needed to make an action decision and execute the action. In addition, the current state of situation awareness at any stage of the takeover process can be represented.

Nevertheless, the model shows limitations. At the moment, productions in the model check static conditions regularly to gather information about the dynamics of the situation. Thus, the model uses several productions to perceive changes. In a next step, productions that are able to trace a moving object could be used for enhancing the model. Another aspect that should be considered in future development is learning. The integration of learning into the model would help represent differences in experience levels between drivers. Furthermore, the model makes a decision and sticks to it. Thus, as soon as a decision is made, the model cannot come back to an earlier decision point. More looping options to change a decision while taking over the driving task have to be implemented.

Moreover, more speed variations of the traffic should be tested and compared to further data of empirical investigations. There might be great differences due to speed deltas that have an influence on the reaction times, as it is in our case with the right complex maneuver. Another aspect, the model does not address yet, are further driver differences in gaze guidance and motoric reactions. Furthermore, the projection stage of situation awareness is not integrated into the model yet. By integrating these points, the model might be able to better predict the deviations that are found in empirical data and represent driver differences for the whole takeover process more closely.

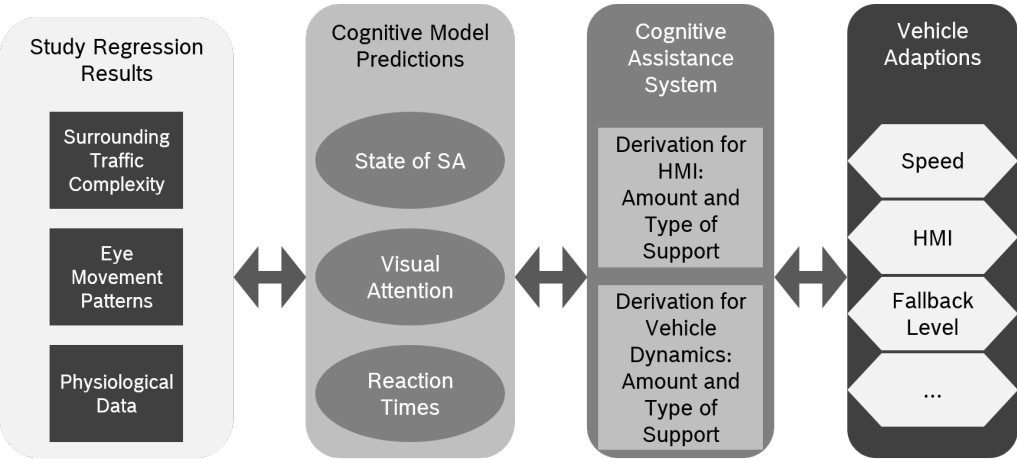

**Figure 8.** Usage of the cognitive model to support situational adaptions of adaptive assistance systems in highly automated driving (source: own figure).

To better predict individual subjects' patterns of performance, the authors of [36] proposed a computational model that accounts for differences in working memory capacity. The integration of this approach into the presented cognitive model could be a first step to further predict individual differences. In combination with that, different forms of learning should be integrated into the cognitive model of the takeover task, as [37] did for a fault-finding task to represent how drivers adapt to the new forms of driving. In the future, more flexible approaches that address the above-mentioned limitations have to be analyzed to better represent driver differences. Especially when dealing with the visual process during a takeover, saccades and fixation times become relevant when predicting the individual takeover process. In a next step, eye-tracking results of a simulator study will be compared to the visual traces of the cognitive model to improve predictions that represent individual variations and analyze errors. Better individualization can result in a cognitive assistance system that is able to guide the driver through a takeover individually and increase safety. Information that the driver needs in a certain situation can be displayed individually and vehicle dynamics can be adapted to hand over the driving task dynamically and adapt the given takeover time individually.

## 5. Conclusions

In order to enhance the usage of such a cognitive model in highly automated driving, some limitations have to be considered. First, the presented model runs only through six scenarios. More scenarios should be tested and compared to a larger amount of empirical data. Second, the productions of the current model check static conditions and compare them to collect information about the dynamics of a situation, but do not actively trace an object. Third, the model has not yet been connected to a driving simulator. These aspects should be met in future research. Furthermore, the model is only able to represent some of the differences between drivers. These are mainly speed adaption and safety habits. Other differences between driver takeovers still have to be investigated that lead to the higher standard deviations that are found in empirical data. However, the model is able to represent different cognitive processes during a takeover in dynamic environments. The usage of

AOIs enhances the simulation of the visual perception. Furthermore, a spectrum of different driving decisions in different situations can be represented. It is able to illustrate the current state of situation awareness in different takeover situations that are found within empirical studies. However, time predictions are close but not perfectly in line with times that are found in empirical data. The model can be used to predict different driving strategies and integrate vehicle and participant data into adaptive assistance systems. By knowing, for example, that a driver is overwhelmed by the situation and has not yet collected enough information of the surrounding traffic environment, the automated system could adapt correspondingly. In such a situation, the system would decelerate, visually guide the driver, and support the maneuver. Using cognitive modeling, assistance systems are able to integrate the cognitive state of the driver on hand and adapt to the situation and the present cognitive state of the driver. Based on such a cognitive model, an automated system could classify the driver's takeover readiness, derive the expected takeover quality [8] and adapt the cognitive assistance for non-critical takeovers accordingly.

**Author Contributions:** Conceptualization, M.S.L.S.-S. and N.R.; methodology, M.S.L.S.-S.; software, M.S.L.S.-S.; validation, M.S.L.S.-S.; formal analysis, M.S.L.S.-S.; investigation, M.S.L.S.-S.; resources, M.S.L.S.-S. and S.W.; data curation, M.S.L.S.-S.; writing—original draft preparation, M.S.L.S.-S.; writing—review and editing, S.W. and N.R.; visualization, M.S.L.S.-S.; supervision, N.R.; project administration, M.S.L.S.-S. All authors have read and agreed to the published version of the manuscript.

**Funding:** This research received no external funding.

**Institutional Review Board Statement:** The study was conducted in accordance with the Declaration of Helsinki, and approved by the or Ethics Committee of TU Berlin (Eingangsdatum des Antrages: 20 March 2019, Antragsnummer: $SF_01_20190320$, Datum der Beschlussfassung: 1 April 2019).

**Informed Consent Statement:** Informed consent was obtained from all subjects involved in the study.

**Data Availability Statement:** Not applicable.

**Acknowledgments:** We wish to acknowledge the help provided by Kathrin Zeeb and Michael Schulz at Robert Bosch GmbH, our department at TU Berlin and Klaus Bengler from TU Munich as second supervisor.

**Conflicts of Interest:** The authors declare no conflict of interest.

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
