# Peer review of "A Cognitive Model to Anticipate Variations of Situation Awareness and Attention for the Takeover in Highly Automated Driving"

_information, doi:10.3390/info13090418_

Round 1

Reviewer 1 Report

Dear Authors, I have read your article with great interest. Your research is very useful. I treat them with great respect. Nevertheless, I had a lot of questions, comments and suggestions.

 Remarks:

1. It is necessary to structure the article material more carefully. The text must be differentiated into subsections.

2. Figure 1 is missing.

3. Obviously, there is an error in Figure 2 – the Follow and Left scripts (Low Compexity versions) are almost identical, which is incorrect. It is necessary to adjust Figure 2 to a more obvious difference between different scenarios.

4. The cognitive model (Fig. 4) is quite realistic, but for the convenience of its perception it would be nice to swap the blue and red blocks identifying movement to the right and left. This wish is connected with the need to increase the spatial visibility of the scheme.

5. Obviously, there is a typo in the text – lines 304-305 "The front is attended (attend-front) and another morotic reaction is executed".

6. Lines 361-366. It would be very correct to present a photo of the experiment on a driving simulator.

7. The phrase "a driving simulator was chosen as driving simulators are identified as the prior testing environment in the field of usability assessments of HMIs and an efficient and risk-free alternative to real driving experiments [30]" (lines 362-364) is formulated incorrectly. Please change it.

8. There is no explanation why the number of participants in the experiment N = 20 was chosen (line 365). Usually, the number of experiments is determined taking into account the variability of the results of preliminary tests. It is also important to indicate those who participated in the experiment - men / women, their age, driving skills, etc.

9. Section 3 – Results. It is important to specify the speed parameters of motion simulation.

If the results of the experiment are estimated in seconds, then it is necessary to know the speed parameters of motion simulation. The delay time of the driver's reaction to a changing situation largely depends on the speed of the car and the transience of changes in the traffic situation. In the article, I did not find data on the speed parameters of motion simulation.

For future research (not in this article, but in other articles), it would be useful for the authors to work out a number of scenarios related to driving into the oncoming lane when overtaking on single-lane (in each direction) roads. You yourself write about this in the conclusion (line 523). That would probably be very interesting.

I wish you success!

Author Response

Reviewer 1

Dear Authors, I have read your article with great interest. Your research is very useful. I treat them with great respect. Nevertheless, I had a lot of questions, comments and suggestions.

Dear Reviewer 1,

Thank you very much for your feedback. We included everything and it really helped improve the paper. In the following, we describe the changes underneath your comments. Thank you again for providing such valuable feedback.

Best, Marlene Scharfe-Scherf

 Remarks:

  1. It is necessary to structure the article material more carefully. The text must be differentiated into subsections.

This is true, we included various subsections.

  1. Figure 1 is missing.

Thank you. We rechecked and recompiled the document. There must have been an issue during the compilation.

  1. Obviously, there is an error in Figure 2 – the Follow and Left scripts (Low Compexity versions) are almost identical, which is incorrect. It is necessary to adjust Figure 2 to a more obvious difference between different scenarios.

Indeed, within the picture the two situations look identical. However, the scenarios must include the lowest possible amount of vehicles, but still trigger the corresponding scenario. Thus, the vehicles also differ in their speed in relation to the ego-vehicle to trigger the scenario. In the following condition, the leading vehicle thus has a higher speed and thus it should be followed, whereas the in the left condition with low complexity the leading vehicle is significantly slower and should thus trigger a lane change to the left. We included this within the text under Scenarios to clarify it. Thank you for the comment.

  1. The cognitive model (Fig. 4) is quite realistic, but for the convenience of its perception it would be nice to swap the blue and red blocks identifying movement to the right and left. This wish is connected with the need to increase the spatial visibility of the scheme.

Thank you for the comment. We increased the special visibility of the scheme to the max that is possible and swapped the colors.

  1. Obviously, there is a typo in the text – lines 304-305 "The front is attended (attend-front) and another morotic reaction is executed".

Corrected, thank you!

  1. Lines 361-366. It would be very correct to present a photo of the experiment on a driving simulator.

True, it is included.

  1. The phrase "a driving simulator was chosen as driving simulators are identified as the prior testing environment in the field of usability assessments of HMIs and an efficient and risk-free alternative to real driving experiments [30]" (lines 362-364) is formulated incorrectly. Please change it.

We changed the sentence, it should make more sense now and the grammar is correct.

  1. There is no explanation why the number of participants in the experiment N = 20 was chosen (line 365). Usually, the number of experiments is determined taking into account the variability of the results of preliminary tests. It is also important to indicate those who participated in the experiment - men / women, their age, driving skills, etc.

We included more information about the participants and as well, information on why we chose the different sample sizes to make the studies comparable. Thank you for the feedback. This improves the paper!

  1. Section 3 – Results. It is important to specify the speed parameters of motion simulation.

If the results of the experiment are estimated in seconds, then it is necessary to know the speed parameters of motion simulation. The delay time of the driver's reaction to a changing situation largely depends on the speed of the car and the transience of changes in the traffic situation. In the article, I did not find data on the speed parameters of motion simulation.

We included information in the results section about the speed parameters within the model as well as within the driving simulator. Both have been synchronized to each other so that the relative speeds within the driving environment are the same. We hope our short explanation clarifies this for you and the readers.

For future research (not in this article, but in other articles), it would be useful for the authors to work out a number of scenarios related to driving into the oncoming lane when overtaking on single-lane (in each direction) roads. You yourself write about this in the conclusion (line 523). That would probably be very interesting.

This is true, great idea!

I wish you success!

Thank you!

Reviewer 2 Report

This paper focuses on a topic of primary importance for developing autonomous vehicles: managing the takeover between autonomous driving and the human driver.

The proposal is based on assessing the human driver situation awareness is considered.

The idea is to analyze the cognitive condition of the driver to aid him/her in getting knowledge about the surrounding environment before put back him/her in the loop. The system itself can become a source of distraction. For this reason, the information proposed is chosen by analyzing the driver's behavior (mainly where he/she looks before takeover the system).

The system does not drive the vehicle but aims to anticipate the driver to takeover better and more safely. These can range from just the visualization of the vehicles surrounding the ego vehicle up to a suggestion of the next manouver to be performed.

It is based on a GUI (unfortunately, the version I received for peer review is missing Figure 1), representing the current situation and eventually suggesting the following driving action to the driver.

The paper is well written and organized. The experimental results are adequately described.

I suggest the following minor revisions:

  • Insertion of Figure 1 depicting the GUI of the system.

Author Response

Reviewer 2

This paper focuses on a topic of primary importance for developing autonomous vehicles: managing the takeover between autonomous driving and the human driver.

The proposal is based on assessing the human driver situation awareness is considered.

The idea is to analyze the cognitive condition of the driver to aid him/her in getting knowledge about the surrounding environment before put back him/her in the loop. The system itself can become a source of distraction. For this reason, the information proposed is chosen by analyzing the driver's behavior (mainly where he/she looks before takeover the system).

The system does not drive the vehicle but aims to anticipate the driver to takeover better and more safely. These can range from just the visualization of the vehicles surrounding the ego vehicle up to a suggestion of the next manouver to be performed.

It is based on a GUI (unfortunately, the version I received for peer review is missing Figure 1), representing the current situation and eventually suggesting the following driving action to the driver.

The paper is well written and organized. The experimental results are adequately described.

I suggest the following minor revisions:

  • Insertion of Figure 1 depicting the GUI of the system.

Dear Reviewer 2,

Thank you very much for you feedback. We highly appreciate it. Concerning figure 1: We rechecked and recompiled the document. There must have been an issue during the compilation, but it is there now. Thank you very much again,

Best, Marlene Scharfe-Scherf

Reviewer 3 Report

This study presents a cognitive model of drivers in takeover situation, in the context of automated driving. The model is compared with empirical data. 

I found the topic of this study very interesting, and I like the way everything is explained and introduced.

More details could have been provided in the description of the empirical study. I actually had to look into the original study to have all the necessary details on the study design and procedure. This information is relevant for estimating statistical differences with the model's results. 

However, I would join the authors in saying that the figures are self-explanatory: the model doesn't really work. As long as the authors honestly admit the limitation of their model, I think this work is still worth publishing.

As a side comment, note that the content of figure 1 is not included in the manuscript. 

Author Response

Reviewer 3:

This study presents a cognitive model of drivers in takeover situation, in the context of automated driving. The model is compared with empirical data.

I found the topic of this study very interesting, and I like the way everything is explained and introduced.

More details could have been provided in the description of the empirical study. I actually had to look into the original study to have all the necessary details on the study design and procedure. This information is relevant for estimating statistical differences with the model's results.

However, I would join the authors in saying that the figures are self-explanatory: the model doesn't really work. As long as the authors honestly admit the limitation of their model, I think this work is still worth publishing.

As a side comment, note that the content of figure 1 is not included in the manuscript.

Dear Reviewer 3,

Thank you very much for your feedback. We highly appreciate it and agree that the model has the limitations that are described within the paper. In future work the model could be developed further to better represent human cognition. We explained the methods section in more detail and rechecked and recompiled the document to make sure that figure 1 is now included. There must have been an issue during the compilation. Thank you again,

Best, Marlene Scharfe-Scherf

Round 2

Reviewer 1 Report

Dear Authors, thank You for your prompt response to my wishes. Now the article has become more qualitative. I think it is worthy of publication in the magazine. I congratulate you on your success!

However, I'm sorry, but Your reaction to the 4th remark is not exactly what I was expecting. It was important not just to change the color of the blocks on the figure 4, but to swap the blocks for greater clarity of the diagram itself. If it's not difficult for You, I still ask you to swap the "Right" - "Left" blocks in places in space. If we consider the scheme of figure 4 from left to right, then the left side in the figure will be on top, not from below.